Role of neuroinflammation mediated potential alterations in adult neurogenesis as a factor for neuropsychiatric symptoms in Post-Acute COVID-19 syndrome—A narrative review

http://orcid.org/0000-0002-9803-270X Saikarthik Jayakumar 1 2 s.jaya@mu.edu.sa
Saraswathi Ilango 3
Alarifi Abdulaziz 4 5
Al-Atram Abdulrahman A. 6
Mickeymaray Suresh 7
Paramasivam Anand 8
Shaikh Saleem 2 9
Jeraud Mathew 10
Alothaim Abdulaziz S. 7
1 Department of Basic Medical Sciences, College of Dentistry, Al Zulfi, Majmaah University , Al-Majmaah, Riyadh , Kingdom of Saudi Arabia
2 Department of Medical Education, College of Dentistry, Al Zulfi, Majmaah University , Al Majmaah, Riyadh , Kingdom of Saudi Arabia
3 Department of Physiology, Madha Medical College and Research Institute , Chennai, Tamil Nadu , India
4 Department of Basic Sciences, College of Science and Health Professions, King Saud Bin Abdulaziz University for Health Sciences , Riyadh , Saudi Arabia
5 King Abdullah International Medical Research Centre , Riyadh , Saudi Arabia
6 Department of Psychiatry, College of Medicine, Majmaah University , Al Majmaah, Riyadh , Kingdom of Saudi Arabia
7 Department of Biology, College of Science, Al Zulfi, Majmaah University , Al Majmaah, Riyadh , Kingdom of Saudi Arabia
8 Department of Physiology, RVS Dental College and Hospital, Kumaran Kottam Campus , Kannampalayan, Coimbatore , Tamilnadu, India
9 Department of Maxillofacial Surgery and Diagnostic Sciences, College of Dentistry, Al Zulfi, Majmaah University , Al Majmaah, Riyadh , Kingdom of Saudi Arabia
10 Department of Physiology, Ibn Sina National College for Medical Studies , Jeddah , Saudi Arabia
Abdullah Jafri
Electronic publication date: 2022 Nov 4
Publication date: 2022
Volume: 10
Electronic Location ID: e14227
Received 2022 Jul 1; Accepted 2022 Sep 22
Copyright: © 2022 Saikarthik et al.
Copyright year: 2022
Copyright holder: Saikarthik et al.
License: This is an open access article distributed under the terms of the Creative Commons Attribution License, which permits unrestricted use, distribution, reproduction and adaptation in any medium and for any purpose provided that it is properly attributed. For attribution, the original author(s), title, publication source (PeerJ) and either DOI or URL of the article must be cited.
License URL: https://creativecommons.org/licenses/by/4.0/

Keywords: Post-acute COVID-19 syndrome, Neuroinflammation, COVID-19, SARS-CoV-2, Neurogenesis, Cytokine storm, Astrocyte, Microglia

Funding: Deanship of Scientific Research, Majmaah University, Kingdom of Saudi Arabia R-2022-298 The authors received support from the Deanship of Scientific Research, Majmaah University, Kingdom of Saudi Arabia for research support under project number: R-2022-298. The funders had no role in study design, data collection and analysis, decision to publish, or preparation of the manuscript.

==============================
Persistence of symptoms beyond the initial 3 to 4 weeks after infection is defined as post-acute COVID-19 syndrome (PACS). A wide range of neuropsychiatric symptoms like anxiety, depression, post-traumatic stress disorder, sleep disorders and cognitive disturbances have been observed in PACS. The review was conducted based on PRISMA-S guidelines for literature search strategy for systematic reviews. A cytokine storm in COVID-19 may cause a breach in the blood brain barrier leading to cytokine and SARS-CoV-2 entry into the brain. This triggers an immune response in the brain by activating microglia, astrocytes, and other immune cells leading to neuroinflammation. Various inflammatory biomarkers like inflammatory cytokines, chemokines, acute phase proteins and adhesion molecules have been implicated in psychiatric disorders and play a major role in the precipitation of neuropsychiatric symptoms. Impaired adult neurogenesis has been linked with a variety of disorders like depression, anxiety, cognitive decline, and dementia. Persistence of neuroinflammation was observed in COVID-19 survivors 3 months after recovery. Chronic neuroinflammation alters adult neurogenesis with pro-inflammatory cytokines supressing anti-inflammatory cytokines and chemokines favouring adult neurogenesis. Based on the prevalence of neuropsychiatric symptoms/disorders in PACS, there is more possibility for a potential impairment in adult neurogenesis in COVID-19 survivors. This narrative review aims to discuss the various neuroinflammatory processes during PACS and its effect on adult neurogenesis.

Introduction

The first case of COVID-19 caused by SARS-COV-2 virus was reported in Wuhan, China on 31st December 2019 and since then the disease has spread to 228 countries throughout the globe (Worldometer, 2022). The incubation period of the SARS-COV-2 virus ranges between 5.1 and 11.5 days with most people developing symptoms after 14 days of active monitoring or quarantine (Lauer et al., 2020). The severity of this disease has a wide range with symptoms like fever, cold, cough, breathing difficulty, pneumonia, other body systems failure and even death has been noted in very severe cases of COVID-19 (WHO, 2022). People with younger age mostly act as asymptomatic carriers whilst the older age group is the most vulnerable group with high severity and mortality (Nuzzo et al., 2021). People with older age (greater than 60 years), pregnancy, chronic pulmonary disease conditions, diabetes and hypertension, cardiovascular diseases and health care workers are high-risk groups for COVID-19 (Ceriello, Stoian & Rizzo, 2020; Huang et al., 2020; WHO, 2022; Wiersinga et al., 2020; Zhou et al., 2020).

Acute COVID-19 has been defined as the period that extends from the onset of symptoms to 3 to 4 weeks. Any symptoms persisting beyond this period are categorized as post-acute COVID-19, where the SARS-COV-2 virus is not detectable (Nalbandian et al., 2021). Similar patterns of persistence of symptoms have been noted previously during the SARS epidemic and MERS outbreak (Ahmed et al., 2020; Hui et al., 2005). A thorough understanding of this phenomenon is vital for the prognosis of the patients as well as to equip healthcare settings to aid in diagnosis and treatment. The post-acute COVID-19 syndrome (PACS) involves multiple organ systems (Nalbandian et al., 2021) and the pathophysiology is held to be different from that of acute COVID-19 (Dixit et al., 2021). Garg et al. (2020) state that PACS is the persistence of symptoms which is sought to be linked with residual inflammation from the convalescent phase of viral replication, organ damage, extended ventilation, or idiopathic (nonspecific) effects of hospitalization. PACS is observed not only in those who had severe forms of COVID-19 but also in outpatients (Montani et al., 2022).

Neuropsychiatric symptoms during the acute stage as well as post-acute COVID-19 are not uncommon ranging from cognitive impairment, delirium, mood changes, and extreme fatigue (Rubin, 2020; Woo et al., 2020). Incidences of dementia, anxiety, and insomnia were noted even after 3 months post-infection (Czeisler et al., 2020). Various studies that assessed the neuropsychiatric symptoms, 14 days to 6 months following acute COVID-19, noted a higher prevalence of symptoms of insomnia, anxiety, depression, and PTSD (Montani et al., 2022). There is conflicting evidence concerning the association between disorders. Several studies showed the relation between the two while several others could not replicate this result (Montani et al., 2022). Though the etiology for such long-lasting effects on the neuropsychiatric facet is still being studied and ever evolving, a few intricate mechanisms have been postulated in the literature. Some of them include biological and environmental factors (Nakamura et al., 2021), virus-induced autoimmunity (Achar & Ghosh, 2020), coagulopathy leading to multi-organ system failure (Achar & Ghosh, 2020), and direct viral infiltration into the nervous system through ACE2 receptor (Gupta et al., 2021; Saikarthik, Saraswathi & Al-Atram, 2021). In an interesting study by Yapici-Eser et al. (2021), it was proposed that SARS-COV-2 proteins mainly the non-structural protein group (NSP) and spike protein mimic various growth factors, such as FGF (1, 2, 4 types), VEGF2, GDNF, IGF, etc. It was hypothesized that such protein mimicking interactions could potentially be associated with neuropsychiatric disorders and variation in risk factors could trigger different pathways presenting with different phenotypes of the disease (Yapici-Eser et al., 2021).

Rationale for the study

Several mechanisms have been proposed previously in the etiology of neuropsychiatric disorders. However, the neuropsychiatric symptoms in PACS and their impact is believed to have long-term consequences which is not much explored. Neuroinflammation has been known to affect cognition, behaviour by means of disrupted BBB, neurotransmission and also by means of impaired neurogenesis (Klein et al., 2021). This narrative review reviews the available literature to address the possible mechanism of COVID-19-induced neuroinflammation as a cause for the various neuropsychiatric symptoms and also to explore the plausible association of impaired neurogenesis in PACS. This timely summary of recent developments would provide a definitive path to researchers, to better understand the pathophysiological basis which would aid in managing the neuropsychiatric symptoms during PACS.

Survey methodology

This study used the narrative review method along with PRISMA-S, which is an extension of PRISMA guidelines for reporting literature search strategies in systematic reviews (Rethlefsen et al., 2021). Due to the scarcity of studies on the effect of COVID-19 on adult neurogenesis, as well as lack of homogeneity in the already published literature, a narrative review style was chosen (Harvey, Schofield & Williden, 2018). Electronic searches were made in databases such as Pubmed, Cochrane, Scopus, Web of Science, Google scholar, and ResearchGate as well as preprint databases such as medRxiv and Research Square. General Google searches were done to report the latest number of COVID-19 cases globally. Being a narrative review, multiple combinations of words were used as search strategies. Some of the words that were used included “SARS-CoV-2”, “COVID-19”, “adult neurogenesis”, “adult hippocampal neurogenesis”, “neuroinflammation”, “hippocampus”, “neuropsychiatric symptoms”, “neuropsychiatric disorders”, “Post-Acute COVID-19 syndrome”, “long COVID-19”, etc. A combination of these words were also used, for example, “neuroinflammation and COVID-19”, “neurogenesis and SARS-CoV-2”. In addition, author names and a list of references were used for search of related references. The last search in the above-mentioned databases was made on 15.06.2022. After reading the abstracts those articles that did not match the requirements of this narrative review were excluded. Articles and preprints in the English language in both clinical and pre-clinical studies were included in this narrative review. Any duplication of articles was removed using the EndNote reference manager (Version 20).

Results

Neuropsychiatric symptoms/disorders in PACS

Survivors of earlier infections caused by other coronaviruses like MERS and SARS presented with an increased risk of neuropsychiatric disorders like anxiety, depression, and PTSD (Hopkins et al., 1999; Rogers et al., 2020). Cognitive decline, decreased mental processing speed, and impairment in memory, attention, and concentration were observed in SARS survivors 1 year after the onset of the disease (Hopkins et al., 1999). A comprehensive systematic review by Rogers et al. (2020) found that out of the 20 neurological and neuropsychiatric complications of COVID-19 that were studied, non-specific symptoms like headache (20.7% (16.1–26.1%)) and anosmia (43.1% (35.2–51.3%)) and core psychiatric disorders of depression (23% (11.8–40.2%)) and anxiety (15.9% (5.6–37.7%)) were found to be highly prevalent. The non-specific symptoms like anosmia, dysgeusia, weakness, and fatigue were the most common, occurring in more than 30% of the patients (Rogers et al., 2020). Many of these complications are capable of becoming a chronic condition and many of the symptoms in PACS could be a continuation of those from the acute phase of the disease (Carfì, Bernabei & Landi, 2020). Survivors of critical illness after discharge from the hospital were found to have a higher prevalence of neuropsychiatric disorders like depression, anxiety, and PTSD (Nikayin et al., 2016; Parker et al., 2015; Rabiee et al., 2016). Most of the neuropsychiatric symptoms of COVID-19 were found to be common in patients with milder forms of the disease (Rogers et al., 2020). Thus, the neuropsychiatric symptoms/disorders are observed in survivors of COVID-19 irrespective of the disease severity which can become chronic.

SARS-CoV-2 entry into the brain

SARS-CoV-2 is a beta coronavirus, a positive sense single stranded RNA virus. Its surface is enveloped with crown-like spikes like other coronaviruses. The spike protein which is responsible for host specificity and tissue tropism is a type-1 glycoprotein. It includes two subunits, S1 for host receptor binding and S2 for the fusion of viral and host cell membrane (Gallagher & Buchmeier, 2001). The cell receptor through which SARS-CoV-2 binds to the host is the ACE2 receptor. The S1 subunit binds with the ACE2 receptor followed by the fusion of S1 to the cell membrane which is mediated by S2. Priming/cleavage of the S1 and S2 subunits is performed by TMPRSS2, a serine protease that is a member of the Hepsin/TMPRSS subfamily (Hoffmann et al., 2020).

SARS-CoV-2-associated central nerve system (CNS) disease has complex and varied pathogenesis. The propensity of the virus to enter the CNS is widely studied. There are three possible routes of viral entry into CNS viz. transmucosal invasion, hematogenous spread, and retrograde neuronal dissemination (Pezzini & Padovani, 2020). SARS-CoV-2 can cross the neural-mucosal interface by infecting the olfactory neurons or diffuse through the channels that are formed by the ensheathing cells of olfactory mucosa and enter the CNS. The virus then may travel along the olfactory tract and reach different areas of the brain connected to it by axonal transport, trans-synaptic transport, or microfusion (Meinhardt et al., 2021; Van Riel, Verdijk & Kuiken, 2015). SARS-CoV-2 can breach the peripheral nerve terminals and can reach the CNS through the trans-synaptic route. It can invade peripheral chemoreceptors and cranial nerves and reach the brain stem (Li, Bai & Hashikawa, 2020). SARS-CoV-2 can also likely enter CNS through gut-brain axis via the enteric nerves (Esposito et al., 2020; Shi et al., 2021). In the hematogenous spread, the virus disseminates the circulation and may breach the blood-brain barrier or blood-CSF barrier to enter the brain or through circumventricular organs that lack blood brain barrier (BBB) (Pezzini & Padovani, 2020). In the Trojan horse mechanism, virus-infected leucocytes may cross the BBB to enter CNS (Desforges et al., 2020) (Fig. 1).

Figure 1 Proposed routes of entry of SARS-CoV-2 into the central nervous system.

Immune response in COVID-19

A cytokine storm is currently considered to be the trademark attribute of the pathogenesis of COVID-19. It is a destructive systemic hyperinflammatory response. It involves autocrine and paracrine activation of various immune cells such as mast cells, macrophages, leucocytes, and endothelial cells which causes increased levels of chemokines and pro-inflammatory cytokines like interleukin-6 (IL-6), IL-1β, IL-8, tissue necrosis factor-alpha (TNF-α), chemokine (C-C-motif) ligand 2 (CCL2), CCL5, IL-17, IL-18, IL-33, CXCL-10, interferon-γ (IFN-γ), and granulocyte-colony stimulating factor (G-CSF) (Azkur et al., 2020; Kempuraj et al., 2020; Li et al., 2020; Nile et al., 2020). SARS-CoV-2 infection activates both immediate and late immune responses in the body. SARS-CoV-2 being a novel coronavirus, there is no prior exposure for the human immune system to this virus and hence it is the innate immune system that acts as the first line of defence (Serrano-Castro et al., 2020). The precise mechanism of immune response to SARS-CoV-2 is not yet fully understood. SARS-CoV-2 which enters the body gets attacked by innate immune cells and the severity of the disease will depend on the capacity of the innate immune system to ward off the virus (Zhu et al., 2020). Coronaviruses are capable of facilitating innate immune suppression and inhibiting adaptive immunity (Oh et al., 2016). Mast cells, macrophages/monocytes, natural killer cells, neutrophils, T lymphocytes, and resident tissue endothelial and epithelial cells are the innate immune cells that get activated by SARS-CoV-2 and are responsible for the cytokine storm in lungs (Azkur et al., 2020; Kempuraj et al., 2020; Kritas et al., 2020). From the initial infection and lysis of the cells (mostly pneumocytes), DAMPs (damage-associated molecular patterns) and PAMPs (pathogen-associated molecular patterns) are produced which activate the innate immune system. DAMPs include cellular contents released from dying cells and proteins released following tissue injury like heat shock protein, heparin sulphate, hyaluronan fragments and PAMPs (pathogen-associated molecular patterns) include oxidized phospholipids and viral RNAs (Imai et al., 2008; Kuipers et al., 2011). These activated immune cells release various pro-inflammatory cytokines, chemokines, proteases, and histamine which help the immune system to fight off the viral infection by recruiting and activating other innate and adaptive immune cells and antiviral gene expression programs (Vardhana & Wolchok, 2020). However, excess activation of these immune cells causes a worsening of the inflammatory response and an increase in the disease severity (Kempuraj et al., 2020). Lymphopenia induced by cytokine storm impairs the adaptive immune system to produce anti-viral antibodies which is critical in the clearance of the virus (Manjili et al., 2020). Cytokine storm and sustained systemic inflammatory response cause acute respiratory distress syndrome (ARDS), multiple organ failure, and death in COVID-19 patients (Li et al., 2020).

An increase in pro-inflammatory Th17 cells and lymphopenia associated with decreased CD4+ T cells, CD8+ T cells, and natural killer cells, and increased cytokine levels (IL-6, IL-10, and TNF-α) were observed in COVID-19 patients (Pedersen & Ho, 2020). The cytokine levels increased during the disease process and declined during the recovery period. Increased levels of IL-6 correlate with mortality and the need for ventilator support (Vardhana & Wolchok, 2020). Patients who are clinically deteriorating were found to present with progressive depletion of lymphocytes while the clinical recovery was preceded by a recovery in lymphocyte count (Chen et al., 2020). The increased levels of IL-6 can further upregulate the cytokine storm in COVID-19 patients. Thus, lymphopenia and the level of cytokine storm are considered to be the markers for COVID-19 which helps to assess and predict disease severity and mortality in COVID-19 patients (Debuc & Smadja, 2021; Kempuraj et al., 2020).

Neurovascular unit and neuroinflammation

Inflammation is the early tissue response to an insult or injury or pathogenic invasion. Neuroinflammation is the inflammatory process in the central nervous system (CNS), which is primarily due to the activation of astrocytes and microglial cells. Astrocytes develop from radial glial cells in due course of neuronal differentiation (Barry & McDermott, 2005) whereas microglia are developed from erythroid-myeloid progenitor cells from fetal yolk sac (Ajami et al., 2007). Microglial cells, the blood-brain barrier (BBB), neurons and the extracellular matrix forms the neurovascular unit (NVU) (Del Zoppo, 2010). The blood-brain barrier varies across each part of the CNS primarily depending on factors such as requirements of the brain region and the diameter of the blood vessel (Rhea & Banks, 2019). The NVU responds to an insult/injury which can lead to disruption of BBB, infiltration of leucocytes, release of inflammatory factors, and activation of microglia & astrocytes (Mracsko & Veltkamp, 2014). Recent studies have shown that microglia and astrocytes exist in a continuum of two extremes as two different phenotypes. Thus, both these cells have pro-inflammatory and anti-inflammatory phenotypes which depend on the signals received by these cells (Jha, Lee & Suk, 2016).

The extracellular and intracellular signals influence the phenotype of microglia. The pro-inflammatory phenotype of microglia (M1) has been known to increase the level of tumor necrosis factor (TNF), IL-1β (interleukin 1 beta), IL-6, and IFN-γ. The release of these inflammatory mediators causes neurotoxicity (by excitotoxicity), neurodegenerative diseases (increased immune activation), and cytotoxicity (release of reactive oxygen species) (Block, Zecca & Hong, 2007; Jha, Lee & Suk, 2016; Smith et al., 2012). On the other hand, the anti-inflammatory phenotype (M2) causes a release of transforming growth factor (TGF), IL-10, IL-13, and IL-4 which provides neuroprotection, a release of trophic factors, and resolution of neuroinflammation (Jha, Lee & Suk, 2016; Orihuela, McPherson & Harry, 2016; Wang et al., 2015). Conditions such as hypoxia or ischemia cause activation of astrocytes called “reactive astrocytes” which has a distinct morphology (Faulkner et al., 2004). These astrocytes release a wide variety of pro-inflammatory and anti-inflammatory cytokines, and chemokines (John, Lee & Brosnan, 2003). Similar to microglia, astrocytes also exist in two phenotypic forms viz. Pro-inflammatory astrocyte (A1) and anti-inflammatory astrocyte (A2) which has diverse effects on the NVU (Fan & Huo, 2021). A1 phenotype secretes pro-inflammatory cytokines such as TNF-α, IL-1β, IL-6, nitric acid, ROS, and glutamate. The A2 phenotype produces neurotrophic factors, thrombospondins, and IL-10 which act as anti-inflammatory mediators (neuroprotective) (Fan & Huo, 2021; Jha, Lee & Suk, 2016).

Neuroinflammation in COVID-19

The cytokine storm in COVID-19 causes disruption of blood-brain barrier and intracranial cytokine storm (Coperchini et al., 2020; Serrano-Castro et al., 2020). Through the disrupted blood-brain barrier, the infiltration of immune cells, and inflammatory cytokines into the brain occurs. This is also one of the pathways of entry of SARS-CoV-2 into the brain. All these activate glial cells, endothelial cells, neurons, mast cells, and other immune cells which trigger neuroinflammatory processes (Coperchini et al., 2020; Kempuraj et al., 2020; Serrano-Castro et al., 2020). SARS-CoV-2 can also enter the cerebral circulation from systemic circulation and attach to ACE2 which is abundant in foot processes of astrocytes, microglia, pericytes, and endothelial cells which are the main cellular element of the blood-brain barrier (Hernández et al., 2021). This process is aided by the sluggish blood flow in cerebral microcirculation, resulting in the disruption of BBB. This in turn will facilitate the entry of SARS-CoV-2 into neurons and glial cells where it can infect and replicate and cause neuroinflammation and neurodegeneration. Thus, SARS-CoV-2 can not only exacerbate pre-existing neuroinflammatory and neurodegenerative conditions but also cause neuroinflammatory and neurodegenerative disorders (Baig et al., 2020).

Another possible mechanism by which the pathologic changes in COVID-19 can cause neuroinflammation could be due to a potential dysregulation of renin angiotensin system (RAS). As mentioned earlier, ACE2 plays a major role in the regulation of RAS. There are two arms/axis in RAS, one is the pro-inflammatory and pro-fibrotic arm, and the other is the anti-inflammatory and anti-fibrotic arm. A variety of proteins and enzymes are involved in the RAS. Angiotensinogen is the precursor that gets converted to Angiotensin-I (Ang-I) by renin. Angiotensin-converting enzyme (ACE) converts Ang-I to Angiotensin-II which acts via AT1 (primary mediator) and AT2 receptors to cause vasoconstriction, increase in vascular permeability, inflammation, angiogenesis, thrombosis, and fibrosis. This arm (ACE/Ang-II/AT1) is the pro-inflammatory and pro-fibrotic arm. ACE2 on the other hand inactivates Ang-II by converting it to its antagonistic peptide, Ang (1-7) which binds with Mas receptors and causes vasodilation, and anti-apoptotic, anti-proliferative, anti-inflammatory effects, and attenuates the signal cascade produced by Ang-II. This arm (ACE2/Ang (1-7)/Mas receptor) is the anti-inflammatory and anti-fibrotic arm (Rice et al., 2004). It is also believed to exhibit anxiolytic and antidepressant effects (de Melo & Almeida-Santos, 2020). A balance in the ACE/ACE2 ratio is critical to maintain an equilibrium between the two arms of RAS. An imbalance in the ACE/ACE2 ratio was implicated in various pathological conditions including Alzheimer’s disease, pulmonary hypertension, cardiovascular, and renal pathology (Bernardi et al., 2012; Kehoe et al., 2016; Lavrentyev & Malik, 2009; Yuan et al., 2015). SARS-CoV-2 induced downregulation of ACE2 depletes the key component of the protective arm of RAS which could result in an unrestrained activation of the deleterious pro-inflammatory and pro-fibrotic arm of RAS. Dysregulation of RAS in the brain is linked with neuroinflammation (Labandeira-Garcia et al., 2017; Rodriguez-Perez et al., 2016). Overactivation of RAS by augmentation of local AT1 receptors was found to exacerbate neuroinflammation (Grammatopoulos et al., 2007; Rodriguez-Pallares et al., 2008; Villar-Cheda et al., 2012).

SARS-COV-2, astrocytes, and microglial interaction

Microglia is a vital innate component of the CNS and astrocytes act as mediators for the SARS-COV-2 infection. Microglia provides anti-viral responses in mild cases and produces neurotoxic effects in severe cases of COVID-19. An increase of pro-inflammatory cytokines caused by both these glial cells can amplify the neuroinflammation and lead to impairment in neurological functions in COVID-19 patients (Vargas et al., 2020). Further, cellular cross-talks between astrocyte, microglia, and endothelial cells are implicated in maintaining the cytokine microenvironment in COVID-19 patients (Matias, Morgado & Gomes, 2019; Vargas et al., 2020). Owing to the important functions of both astrocyte and microglia in homeostasis and during viral episodes, it is highly possible for the involvement of these cells in the post-acute phase of SARS-COV-2 infection.

Adult neurogenesis

There are several contradictory pieces of evidence on adult neurogenesis in humans mainly spurred by the lack of direct evidence from live human subjects (Berger, Lee & Thuret, 2020).

The formation of new neurons in the adult brain from neural stem cells and neural progenitor stem cells is called neurogenesis. During embryonic development, it is involved in the formation of the brain, and in the adult brain, it persists in certain areas of specialized microenvironment called the neurogenic niche. The neurogenic niche plays a crucial role in the maintenance and regulation of neural stem cell proliferation and contains various trophic factors, hormones, vasculature, and glial cells that enhance neurogenesis (Mu, Lee & Gage, 2010). New neurons are generated by the neurogenic niche throughout adult life in response to both physiological and pathological stimuli (Fan & Pang, 2017). Neurogenesis involves the generation of new neurons, glial cells, oligodendrocytes, and astrocytes. It is a complex process that includes cellular proliferation, differentiation, survival, and integration. There are numerous intrinsic and extrinsic factors that regulate neurogenesis in an integrated manner. The events in neurogenesis occur in two phases, the early phase of proliferation, fate commitment, and cellular migration and the late phase of development of synaptic circuitry and survival of the neurons (Pathania, Yan & Bordey, 2010). The subventricular zone (SVZ) of lateral ventricles and the subgranular zone (SGZ) of the dentate gyrus of the hippocampus are the two sites of adult neurogenesis (Toda et al., 2019).

The stem cells in the subgranular zones get differentiated into neural progenitor cells which become immature neurons and then mature neurons. However, only 15–30% of immature neurons survive the maturation process. The survived mature neurons become granule cells whose axons form the mossy fibers extending to the hilus and CA3 region and their dendrites in the molecular layer receive connections from the entorhinal cortex. Over a period of several weeks, they show increased synaptic plasticity and become indistinguishable from other older granule cells (Kempermann, Song & Gage, 2015). Newborn neurons at the subventricular zone migrate to the striatum and they differentiate to form striatal interneurons and in rodents, they migrate along the rostral migratory stream (RMS) and differentiate to form interneurons of the olfactory bulb (Shohayeb et al., 2018). Neurogenesis in the hippocampus is a unique form of brain plasticity that plays a crucial role in memory, learning, pattern separation, and cognitive flexibility.

Dysregulation of adult neurogenesis in the hippocampus is associated with psychiatric symptoms and cognitive decline in psychiatric and neurological disorders (Toda et al., 2019). In addition to the role of neurogenesis in physiological conditions, the newly generated neurons also move to sites of brain injury and form the endogenous repair system (Apple, Fonseca & Kokovay, 2017). It has been found that apart from SGZ and SVZ, certain areas of the adult brain like the neocortex, tegmentum, substantia nigra, amygdala, brainstem, and spinal cord also retain some neurogenic potential. However, more explorations are needed to confirm this and elucidate the functional significance (Fan & Pang, 2017) (Fig. 2).

Figure 2 Adult neurogenesis in the dentate gyrus of hippocampus.

NSC, neural stem cell; NB, neuroblast; IN, immature neuron; MN, mature neuron; CA1 and CA3, Cornu Ammonis 1 and 3 regions.

Why adult neurogenesis is important in relation to COVID-19 and PACS?

Research studying adult neurogenesis in COVID-19 and COVID-19 survivors is scarce. However, extrapolation of the results of some recent studies allows for a speculation that adult neurogenesis can have a role to play in the neuropsychiatric symptoms/disorders in COVID-19 and PACS.

Firstly, neuropsychiatric disorders like depression, anxiety, and PTSD are found to be prevalent in COVID-19 survivors (Rogers et al., 2020; Tu et al., 2021). Recent studies postulate a potentially increased risk of developing and/or worsening existing neurodegenerative disorders like Alzheimer’s disease and Parkinson’s disease in COVID-19 patients (Brundin, Nath & Beckham, 2020; Ciaccio et al., 2021; Leta et al., 2021; Sulzer et al., 2020). One of the common features of these neuropsychiatric conditions is that they correlate well with cognitive deficits, mood dysregulation, and a reduction in hippocampal volume and they display impaired adult neurogenesis (DeCarolis & Eisch, 2010).

Secondly, brain imaging studies revealed a negative correlation between hippocampal grey matter volume and loss of memory, and severity of post-traumatic stress syndrome (PTSS) in COVID-19 survivors (Lu et al., 2020; Tu et al., 2021). Memory acquisition depends on newborn neurons and a decrease in adult hippocampal neurogenesis is implicated in the impairment of acquisition of memory (Misane et al., 2013; Recinto et al., 2012). Though the studies found an increase in gray matter volume in COVID-19 survivors 3 months and 1 year after their recovery, it could be attributed to the ongoing nature of the traumatic event of the pandemic with elevated levels of stress and anxiety and a compensatory response (Lu et al., 2020; Tu et al., 2021).

Thirdly, anosmia is a key feature of acute COVID-19 and is also observed in PACS (Araújo, Arata & Figueiredo, 2021; Aziz et al., 2021). Recent brain imaging studies show dysfunction, abnormalities, and atrophy of the olfactory bulb in COVID-19 patients and patients suffering from PACS who presented with anosmia (Chiu et al., 2021; Galougahi et al., 2020; Kandemirli et al., 2021). Neurogenesis in the olfactory epithelium and olfactory bulb is essential for the sense of smell and anosmia is associated with impaired adult olfactory neurogenesis (Boesveldt et al., 2017; Lledo & Valley, 2016). In addition, anosmia is an important pre-motor symptom of Parkinson’s disease which appears to have no direct association with the neurodegenerative process of substantia nigra but seems to be related to impaired adult neurogenesis (Marxreiter, Regensburger & Winkler, 2013; Winner, Kohl & Gage, 2011). COVID-19 is theorized to cause defects in the dopamine system, loss of dopaminergic neurons, and an exacerbation of clinical features of Parkinson’s disease (Brundin, Nath & Beckham, 2020; Sulzer et al., 2020).

Finally, the role of ACE2 in adult neurogenesis in COVID-19 gives a much more vital perspective on the discussion at hand. ACE2 is a surface membrane protein that acts as an obligatory receptor for SARS-CoV-2 and facilitates its entry into the host cell (Hoffmann et al., 2020). In addition to serving as a receptor for SARS-CoV and SARS-CoV-2 virus, ACE2 also acts as a negative regulator of the renin-angiotensin system (RAS) and facilitates amino acid transport in the intestine (Gheblawi et al., 2020; Hoffmann et al., 2020). Various experiments conducted on rodent models give insight into the sites of ACE2 expression. ACE2 is expressed mainly in the lungs, intestine, brain, liver, heart, kidney, and testes. In the brain, it is expressed in neurons, oligodendrocytes, and astrocytes and the sites of ACE2 expression in the brain include ventricles, hippocampus, hypothalamus, substantia nigra, middle temporal gyrus, pontine nuclei viz. pre-Bötzinger complex and nucleus of tractus solitarius and in the olfactory bulb (Gheblawi et al., 2020). More importantly, ACE2 is highly expressed in the key components of the blood-brain barrier viz. astrocytes, astrocytic foot processes, pericytes, and endothelial cells (Hernández et al., 2021).

A recent study conducted using human induced pluripotent stem cells (iPSC) derived neural cells found ACE2 expression in young neurons and human-induced pluripotent stem cell-derived neural progenitor cells (Kase & Okano, 2020). The tissues and organs that are the major target sites for SARS-CoV-2 are those which has higher expression of ACE2 (Pagliaro & Penna, 2020). Similar to SARS-CoV, binding of SARS-CoV-2 with ACE2 causes downregulation of ACE2 (Datta et al., 2020; Seltzer, 2020; Tang et al., 2021; Triana et al., 2021). This downregulation of ACE2 will cause dysregulation of RAS and other complications in addition to its direct effects. Out of the many physiological functions of ACE2, its neuroprotective role is of prime importance to this discussion. Pre-clinical experiments conducted in animal models show the diverse neuroprotective function of ACE2. In an Alzheimer’s disease rodent model, Diminazene, an ACE2 activator was found to increase CREB, BDNF, and nicotinic receptors while reducing apoptotic and inflammatory proteins which all play a major role in adult neurogenesis (Kamel et al., 2018). In transgenic mice, neurotoxic amyloid protein Aβ43 is converted to a neuroprotective form Aβ40 by ACE2 (Liu et al., 2014). ACE2 deficient mice exhibited impaired memory and learning, and abolition of exercise-induced adult hippocampal neurogenesis (Klempin et al., 2018; Wang et al., 2016).

ACE2 is involved in the intestinal neutral amino acid transport via the neutral amino acid transporter BoAT1. ACE2/BoAT1 complex regulates the gut microbiota composition and function. ACE2 knock-out animals presented with impaired gut microbiota composition (Hashimoto et al., 2012). SARS-CoV-2 entry into the enteric host cells leads to ACE2 shedding by S priming which may lead to gut microbiota dysbiosis (He et al., 2020; Viana, Nunes & Reis, 2020). There is an increase in interest among the researchers regarding a potential link between gut microbiota and the development of neuropsychiatric disorders linked to impaired adult neurogenesis like anxiety and depression (Peirce & Alviña, 2019). Prolonged antibiotic treatment-induced depletion of gut microbiota in adult mice caused an impairment in adult neurogenesis and cognitive function (Möhle et al., 2016). Thus, gut microbiota dysbiosis could be another way through which ACE2 downregulation by SARS-CoV-2 may lead to impaired adult neurogenesis.

ACE2 is involved in the intestinal absorption of tryptophan, the precursor of serotonin which plays a major role in adult neurogenesis and is implicated in psychiatric illness like anxiety and depression. Downregulation of ACE2 reduces serotonin levels in brain thereby affecting adult neurogenesis (Klempin et al., 2013).

Hence, based on the above-mentioned factors, COVID-19 may have a potential impact on adult neurogenesis which could be implicated in the neuropsychiatric symptoms/disorders in COVID-19 survivors. The current review is speculative and relied on thorough literature review discusses the possible implication of potentially impaired adult neurogenesis in neuropsychiatric symptoms/disorders in PACS with emphasis on the role of neuroinflammation.

Neuroinflammation and hippocampus in PACS

It has been elucidated recently that prolonged inflammation caused by a release of pro-inflammatory cytokines can cause some neurological deficits and cognitive dysfunction during the post-acute phase of COVID-19 (Maltezou, Pavli & Tsakris, 2021). Recent studies point towards the persistence of neuroinflammation in patients 3 months after recovery from COVID-19 which emphasize the link to the neuropsychiatric sequelae of COVID-19 in PACS (Goldberg et al., 2021; Lu et al., 2020). A recent study by Serrano-Castro et al. (2022) found that the chemokine and growth factor profile of COVID-19 patients, 3 months after discharge depicted a persistent neuroinflammatory state.

Researchers across the globe use different small and large animal models to study COVID-19 and PACS regarding host response, transmission, pathogenesis, and therapeutic strategies. The World Health Organization (WHO) has assembled WHO-COM (WHO COVID-19 modelling), an international panel to develop and study new animal models for COVID-19 research. Readers can refer to the review by Muñoz-Fontela et al. (2020) for information regarding animal models used in COVID-19 research. Various viral infections were found to affect hippocampal functioning including neurogenesis, protein and neurotrophin expression, neuron morphology and function (Bobermin et al., 2020; Francesca et al., 2006; Hosseini et al., 2018; Li Puma et al., 2019; von Rüden et al., 2012). SARS-CoV virus-infected C57/BL6 mice model showed that viral RNA and the live virus could be isolated from the brain of infected mice which was mainly localized in the hippocampus (Glass et al., 2004). A recent study by Klein et al. (2021) found that SARS-CoV-2 infected hamsters and a post-mortem study of brains of patients deceased from COVID-19 showed disruption in BBB, activation of microglia and, increased expression of brain-derived IL-1β and IL-6 in the hippocampus and lower medulla. The study also concluded that the persistence of neurological problems as noted in PACS could be mediated due to neuroinflammation affecting neural vasculature, neurotransmission and neurogenesis (Klein et al., 2021).

Neuroimaging studies in live patients (Chiveri et al., 2021; Moriguchi et al., 2020) and post-mortem brain studies (Fabbri et al., 2021; Solomon et al., 2020; Thakur et al., 2021) have shown neuropathogenic changes in the hippocampus caused by SARS-CoV-2 infection. Given the implication of hippocampal pathology in various neuropsychiatric disorders, SARS-CoV-2 mediated neuropathogenic changes in the hippocampus could be attributed to neuropsychiatric disorders like depression in PACS (Nestler et al., 2002; Roddy & O’Keane, 2019; Roddy et al., 2019). SARS-CoV-2 induced potential impaired adult hippocampal neurogenesis could very well be one of the underlying cellular mechanisms behind neuropsychiatric symptoms/disorders in COVID-19 survivors. Future studies to elucidate the role of SARS-CoV-2-induced neuroinflammation and a possible impairment in adult neurogenesis in the development of neuropsychiatric disorders are much needed.

Putative role of neuroinflammation in potentially impaired adult neurogenesis in PACS

Earlier studies have shown that the hippocampus is highly susceptible to the effects of neuroinflammation (Barrientos et al., 2015; Hueston et al., 2018). The expression of IL-1β, a pro-inflammatory cytokine that is an important mediator of neuroinflammation, and its receptor are at high levels in the hippocampus (Ban et al., 1991; Parnet et al., 1994). Acute exposure to IL-1β disrupts adult hippocampal neurogenesis and contributes to cognitive and memory impairments in stress-related psychiatric disorders (McPherson, Aoyama & Harry, 2011; Ryan et al., 2013). Chronic exposure to IL-1β causes impairment in adult hippocampal neurogenesis which affects hippocampal-dependent processes like pattern separation (Hueston et al., 2018). There are different mechanisms by which neuroinflammation affects adult neurogenesis as discussed below (Fig. 3).

Figure 3 Putative mechanism depicting the effect of neuroinflammation on adult neurogenesis during PACS.

The entry of SARS-COV-2 virus into the brain triggers the release of proinflammatory cytokines which may potentially affect the hippocampal neurogenesis. This could be possibly hypothesized as the reason for the various neuropsychiatric symptoms that are present during PACS.

Glial cells

In normal physiological conditions, the neuroglial pathways and network operate to maintain neuronal health and circuitry. In the case of chronic inflammatory conditions, there occurs an imbalance in the cytokines in the microenvironment which activates neurodegenerative pathways (Yap et al., 2021; Zhang, Zhang & You, 2018). One of the ubiquitous element of neuroinflammation is the activation of astrocytes and microglia (Glass et al., 2010; Tjalkens, Popichak & Kirkley, 2017). They affect neurogenesis by the secretion of inflammatory mediators.

Microglia

Microglia secretes the growth factors, brain-derived neurotrophic factors (BDNF) and insulin-like growth factors (IGF-1) which play a key role in adult hippocampal neurogenesis (Nakajima et al., 2001; Suh et al., 2013). Experimental evidence shows that these factors are expressed in the regions of SGZ and hippocampus during adulthood, though found to be decreased initially after birth (Dyer et al., 2016; García-Segura et al., 1991; Mori, Shimizu & Hayashi, 2004). Inhibition of neural progenitor cell proliferation and reduction in the thickness of granule cells was noted in BDNF receptor, TrkB knockout mouse (Galvão, Garcia-Verdugo & Alvarez-Buylla, 2008). IGF-1 promotes neural precursor cell (NPC) proliferation, differentiation as well as survival probably by anti-apoptotic effects (Åberg et al., 2003). It was found that voluntary exercise increased neurogenesis by increasing the proportion of microglia that expresses BDNF and IGF-1 (Kohman et al., 2012; Littlefield et al., 2015). Short-term signaling of these neurotrophic factors viz., BDNF, and IGF, mediates cellular plasticity needed for learning and memory, whereas long-term signalling leads to neurogenesis (Duman, 2004).

Microglia are similar to macrophages and are primarily responsible for maintaining brain homeostasis and response to injury (Block, Zecca & Hong, 2007). Activated microglia become ameboid-shaped and express ACE2 and transmembrane protease serine subtype 2 (TMPRSS2) (Singh, Bansal & Feschotte, 2020). A recent study has shown that microglia are directly infected by the SARS-COV-2 virus and can cause self-apoptosis, thereby causing a reduction in the number of microglia which leads to further infiltration of the virus (Jeong et al., 2022). Secondly, infection with SARS-COV-2 significantly increased the level of TNF-α and IL-6, suggesting that activated microglia lead to neuroinflammation (Jeong et al., 2022). A Post-mortem study of brains of patients deceased from COVID-19 showed neuropathological signs of microglial activation (Matschke et al., 2020). A study by Huang et al. (2020) showed that plasma levels of pro-inflammatory markers including different types of IL, FGF, IFN-γ, TNF-α, and VEGF were increased in severe COVID-19 patients who needed admission to intensive care unit. It could be hypothesized that the release of these inflammatory cytokines by activated microglia could lead to the breakage of BBB precipitating various neurological signs and deficits in COVID-19 infected patients (Vargas et al., 2020). Very recently, a “two-hit” hypothesis of activation of microglia has been proposed, which could explain the vulnerability of certain groups (aging, co-morbidity, poor diet) for severe COVID-19 infection and prolonged sickness behavior (Bouayed & Bohn, 2021).

Astrocytes

In 2002, Song, Stevens & Gage (2002) discovered that in adult rats, astrocytes promote neural precursor cell differentiation to neurons in the hippocampus but not in the spinal cord. BDNF, fibroblast growth factor 2 (FGF-2), glial cell-derived neurotrophic factors (GDNF) and vascular endothelial growth factors (VEGF) are the neurogenic growth factors secreted by astrocytes (Araki, Ikegaya & Koyama, 2021). Astrocytic BDNF acts on the post synaptic cells of the hippocampus and stimulates neurogenesis. Such activity was found to alleviate anxiety-like symptoms in experimental mice (Quesseveur et al., 2013). Acute stress potentiates hippocampal neurogenesis that was mediated through astrocyte secreted FGF-2 and neutralizing FGF-2 prevented the proliferation of NPCs in cultures (Kirby et al., 2013). Dexmedetomidine was found to mediate neurogenesis in Dentate gyrus (DG), by upregulating the expression of GDNF derived from astrocytes, neural cell adhesion molecule (NCAM) and cAMP response element-binding protein (CREB) by improving astrogenesis (Zhang et al., 2019). In a recent study, it was found that enhanced VEGF promoted neurogenesis by transdifferentiation of astrocytes to neurons and such effects were abolished after treatment with Flurocitrate which is an astrocyte inhibitor in the striatum of the ischemic stroke model (Shen et al., 2016).

Owing to the importance of astrocytes in the formation of BBB, it could be postulated that infection of astrocytes with SARS-COV-2 virus could compromise the integrity of BBB (DeOre et al., 2021). Previously, compromise in BBB, and neuroinflammation have been implicated in various neurodegenerative and neuropsychiatric disorders caused due to several types of Viral infections (Palus et al., 2017; Persidsky et al., 2000; Verma et al., 2010). Conversely, disrupted BBB could in turn activate astrocyte and microglial cells as an innate immune response (Alquisiras-Burgos et al., 2021). Elevated levels of glial fibrillary acidic protein (GFAP) were noted in COVID-19 patients which is a marker for astrogliosis (Heimfarth et al., 2022). As astrocytes are principal producers of cytokines and chemokines in natural immune response, it could be held that they can cause neuroinflammation and neurotoxicity after infection (Tavčar et al., 2021) and also serve as a host for viral replication (Crunfli et al., 2021). A post-mortem study conducted on the brains of COVID-19 patients showed astrocytes to be the major site of infection and replication of SARS-CoV-2 (Crunfli et al., 2021).

Pro-inflammatory cytokines

The pro-inflammatory cytokines in the brain are mainly produced by activated microglia (Wang & Jin, 2015). Depending on the physiological state, the action of cytokines in the regulation of adult neurogenesis varies. Under physiological conditions, IL-6 and TNF-α activate neurotrophic factors and promote neuroregeneration and IL-2 participates in BDNF signaling and hippocampal functioning. However, in a proinflammatory environment, the action of these cytokines leans more towards neurodegeneration and is implicated in the pathogenesis of neuropsychological disorders (Baune et al., 2012; Beck et al., 2005; Eker et al., 2014; Murphy et al., 2000). Chronic neuroinflammation directly impairs adult hippocampal neurogenesis though there are controversial results (Fan & Pang, 2017). Proinflammatory cytokine IL-1β, IL-6, and IFN-α causes a reduction in neural cell proliferation and suppresses adult hippocampal neurogenesis (Borsini et al., 2017; Borsini et al., 2018; Koo & Duman, 2008). TNF-α has a dual effect on adult neurogenesis in vivo. TNFR1 receptor activation causes suppression of neurogenesis while TNFR2 activation favors neurogenesis. I. vitro effect of TNF-α was predominantly suppressive to adult neurogenesis (Chen & Palmer, 2013). A dose-dependent inhibition of adult neurogenesis was produced by overexpression of IL-1β (Wu et al., 2012). Nuclear factor-Kβ signaling is found to be the mediator for the anti-neurogenic effect of IL-1β (Koo et al., 2010). Chronic expression of IL-1β in DG both in vitro and in vivo resulted in a reduction in hippocampal neurogenesis (Mathieu et al., 2010). IL-6 is considered to be the pivotal xcytokine that inhibits adult neurogenesis (Wang & Jin, 2015). IL-6 impairs neurogenesis by promoting NPCs towards gliogenesis (Vallieres et al., 2002). Chronic overexpression of IL-6 in astroglia causes a significant reduction in new neuron production without affecting gliogenesis (Vallieres et al., 2002). Neural stem cells exposed to IL-6 and TNF-α exhibited a marked reduction in neurogenesis (Monje, Toda & Palmer, 2003). There is conflicting evidence on the in vitro effect of IFN-γ on adult neurogenesis (Wang & Jin, 2015). However, in vivo studies show insignificant neurogenesis suppressing effect by IFN-γ (Monje, Toda & Palmer, 2003).

Neurotrophic factors like brain-derived neurotrophic factor (BDNF), insulin-like growth factor-1 (IGF-1), nerve growth factor (NGF), glia-derived nerve factor (GDNF), fibroblast growth factor 2 (FGF-2), and epidermal growth factor (EGF) play a key role in the regulation of adult neurogenesis (Saikarthik, Saraswathi & Al-Atram, 2021).

There exists an inverse relationship between BDNF and pro-inflammatory cytokines, IL-6, IL-2, TNF-α, INF-γ, IL-1β in pro-inflammatory states (Yap et al., 2021). IL-1β was shown to inhibit the neuronal expression of BDNF in the presence of glial cells (Rage, Silhol & Tapia-Arancibia, 2006). Inflammatory cytokines interfere with BDNF signaling by influencing TrkB phosphorylation (Cortese et al., 2011). Administration of IFN-α causes a reduction in BDNF levels (Lotrich, Albusaysi & Ferrell, 2013). Chronic neuroinflammation causes a reduction in the microglial release of neurotrophic factors like IGF-1, thereby causing neurodegeneration (Labandeira-Garcia et al., 2017; Suh et al., 2013). TNF-α inhibits IGF-1 signalling in neurons (Venters et al., 1999).

Serum BDNF levels were found to be decreased in COVID-19-positive patients and were found to be restored during recovery (Azoulay et al., 2020). No significant difference was noted in the levels of IGF between COVID-19 positive and normal patients. However increased levels of IGF were associated with hypertension, neurogenic disease and shock which were noted in severe cases of COVID-19 (Feizollahi et al., 2022). Thus, the role of BDNF and IGF is found to be, and hence further studies are necessary to study the effect of these neurotrophic factors on neurogenesis in the post-acute COVID-19 phase.

Anti-inflammatory cytokines and chemokines

A wide range of actions on adult neurogenesis is demonstrated by anti-inflammatory cytokines and chemokines. Anti-inflammatory cytokines IL-4, IL-10 that are released during neuroinflammation promote neurogenesis. In COVID-19, IL-10 levels are increased which promotes neuronal migration (Butovsky et al., 2006; Lorkiewicz & Waszkiewicz, 2021). Increased expression of TGF-β was observed in COVID-19 patients which has pro-neurogenic effects (Samsami et al., 2022; Xiong et al., 2020). Chronic expression of TGF-β improves adult hippocampal neurogenesis (Mathieu, Piantanida & Pitossi, 2010). A recent study found that chemokines viz. stromal cell-derived factor–1 (SDF-1) and monocyte chemoattractant protein-1 (MCP-1) levels to be higher in COVID-19 patients 3 months after their hospital discharge (Serrano-Castro et al., 2022). They are released by astrocytes and their levels are upregulated during neuroinflammatory states. The receptors of SDF-1a, an isoform of SDF-1 viz. CXCR4 and CXCR7 and the receptor for MCP-1, CCR2 are highly expressed in NSCs (Ni et al., 2004; Peng et al., 2004; Widera et al., 2004). Both these chemokines play a major role in the migration of NSCs during neurogenesis. They also were shown to play a positive role in neuronal proliferation and differentiation (Lee et al., 2013; Wu et al., 2009). Mildly symptomatic and severe cases of COVID-19 presented with higher levels of fractalkine (Khalil, Elemam & Maghazachi, 2021). Neuronal CX3CL-1 (fractalkine)/CX3CR1 signalling has a regulatory role in adult neurogenesis with disruption in the signalling causing decreased survival and proliferation of NPCs in rodent model (Bachstetter et al., 2011). CCL11 (eotaxin-1) which acts through receptor CCR3 was found to be increased in the earlier phase of COVID-19 and its levels remained steady post infection (Khalil, Elemam & Maghazachi, 2021). Increased levels of peripheral CCL11 decreased adult neurogenesis and affected learning and memory in animal model (Villeda et al., 2011). A predominantly positive impact of neuroinflammation on adult neurogenesis is exhibited through anti-inflammatory cytokines and chemokines.

Conclusion

From the above discussion, we could postulate that neuroinflammation in PACS has the potential to cause alterations in adult neurogenesis. COVID-19 worsens pre-existing neuroinflammatory and neurodegenerative conditions like major depressive disorder, Alzheimer’s disease, and Parkinson’s disease in addition to causing new such conditions. Some of the features of PACS including depression, memory loss, and cognitive disorder has been associated with impaired adult neurogenesis. With neuroinflammation having both beneficial and detrimental effects on neurogenesis, based on the prevalence of neuropsychiatric symptoms in PACS, the detrimental effects seem to outweigh the beneficial ones. Hence, impairment in adult neurogenesis can be a potential cause for the neuropsychiatric symptoms/disorders in PACS. However, preclinical studies to specifically analyze adult neurogenesis in SARS-CoV-2 infection are crucial. A better comprehension of the process of adult neurogenesis in PACS may help elucidate the potential role of the regenerative capacity of neural precursor cells and adult neurogenesis in battling the neuropsychiatric symptoms/disorders in PACS. Targeted therapeutic strategies to manage neuroinflammation and impaired adult neurogenesis are the need of the hour to prevent the development of neurological complications of PACS.

Additional Information and Declarations

Competing Interests

Author Contributions

Data Availability

Jayakumar Saikarthik is an Academic Editor for PeerJ.

Jayakumar Saikarthik conceived and designed the experiments, analyzed the data, prepared figures and/or tables, and approved the final draft.

Ilango Saraswathi conceived and designed the experiments, analyzed the data, prepared figures and/or tables, authored or reviewed drafts of the article, and approved the final draft.

Abdulaziz Alarifi conceived and designed the experiments, prepared figures and/or tables, and approved the final draft.

Abdulrahman A. Al-atram conceived and designed the experiments, prepared figures and/or tables, and approved the final draft.

Suresh Mickeymaray performed the experiments, authored or reviewed drafts of the article, and approved the final draft.

Anand Paramasivam performed the experiments, authored or reviewed drafts of the article, and approved the final draft.

Saleem Shaikh performed the experiments, prepared figures and/or tables, authored or reviewed drafts of the article, and approved the final draft.

Mathew Jeraud conceived and designed the experiments, performed the experiments, analyzed the data, prepared figures and/or tables, and approved the final draft.

Abdulaziz S. Alothaim conceived and designed the experiments, analyzed the data, prepared figures and/or tables, authored or reviewed drafts of the article, and approved the final draft.

The following information was supplied regarding data availability:

This is a literature review and there is no raw data.

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
