# Peer review of "Role of neuroinflammation mediated potential alterations in adult neurogenesis as a factor for neuropsychiatric symptoms in Post-Acute COVID-19 syndrome—A narrative review"

_PeerJ, doi:10.7717/peerj.14227_

## Round 0.1 · original submission · Major Revisions

Please do the necessary major revisions, thank you.

Reviewer 1 ·

Basic reporting

This review is well-written, although could benefit from more detailed and informative figures and tables. It is not broad in its application, being more specific for the fields of psychiatry and neuroscience, both basic and clinical. However, this is fine as it is aimed at said audience. The review falls within the scope of the journal. The introduction is adequate.

There have been a great many reviews covering the neuropsychiatric symptoms of post acute Covid-19 syndrome (PACS) in the past few months, many of which also consider the possible neurobiological basis for the condition, including inflammation. Considering this, the authors should elaborate more in the Introduction, with detailed account in subsequent sections, on how their review addresses something unique or extends the current literature. This is elaborated on below.

Under “Rationale for the Study” the authors state that their review is timely to “provide a definitive path to researchers to work on targeted treatment strategies”, yet none of these issues are addressed in any great depth. So for instance in the Conclusion the authors note that pre-clinical studies describing how SARS-CoV-2 affects neurogenesis is needed, and here the pre-clinical underpinnings for this statement needs to be built out somewhat (see below). The conclusion also notes that specific targeted therapeutic strategies are needed, and perhaps here the authors could provide a table that elaborates more on these targets for future reference and directing research, especially what’s been done etc. By implication, relevant detail on this needs to be presented in the review perhaps under a separate heading (see below).

The section on “Antiinflammatory cytokines and chemokines” (line 618) is very brief and could be expanded on. In this regard, specific attention can be made here to how the authors see the use of anti-inflammatory drugs in treating PACS. Are there some with greater or lesser pharmacological attributes that clinicians could consider useful in treating PACS, if at all, e.g. COX II inhibitors, N acetyl cysteine and so on? Further on this point, are their Covid-19 specific inflammatory pathways that lend themselves to better targeting by certain anti-inflammatory compounds?

The authors describe how ACE2 is involved in Covid-19 infection and later inflammation yet don’t elaborate on the role of ACE2 in psychiatric illness, per se. This is important as modulators of the angiotensin system present with antidepressant properties. Would these drugs target inflammation and hence offer additional clinical utility in treating PACS?

Another inflammatory target could be the nitric oxide-cGMP system, which is also implicated in various neuropsychiatric disorders. Would drugs that target these processes have therapeutic potential within the scope outlined by this review?

The authors briefly mention how dexmedetomidine may modulate neurogenesis, yet don’t elaborate on its primary mode of action (lines 548-551), viz. an alpha2A agonist. Again, these actions have implications in treating neuropsychiatric diseases. Does this connect to neuroplastic and neuroinflammatory actions in PACS?

Another important issue that is lacking is whether known drugs used to treat the typical neuropsychiatric manifestations of PACS, including antidepressants, anxiolytics, antipsychotics etc, modify immune-inflammatory responses, especially the inflammatory processes noted in this review. Furthermore, would the authors say that such drugs offer broad utility to treat covid-19 related inflammation plus the underlying neurobiology of the presenting mood/anxiety disorder? This suggests a win-win situation but needs deeper discussion. Indeed, psychotropic drugs have profound effects on inflammatory markers, which can be touched on in the section on “Antiinflammatory cytokines and chemokines”.

The authors make brief mention of animal models, e.g. SARS-CoV-infected C57/Bl-6 mice. A distinct shortcoming in Covid-19 research is the availability of suitable translational animal models. Since this review attempts to “provide a definitive path to researchers to work on targeted treatment strategies” for PACS, an animal model of PACS is especially critical given that this condition may well be around for a while. Can the authors offer any guidance on this? Are there long-covid models out there and if not, what are the current drawbacks preventing their development and application? Such detail will add significant impact to this review.

In lines 534-537, the authors mention the use of dual-hit models to address questions pertaining to microglia activation, inflammation and vulnerability to develop neuropsychiatric illness. This is a commendable idea, especially one that has relevance for understanding how PACS associated inflammation may prompt the development of treatment resistance in patients with a history of psychiatric illness, e.g. treatment resistant depression. This may represent a direction for future Covid research and warrants deeper discussion from the literature.

Please look at lines 369-371, which needs re-writing for clarity.

Please look at lines 430-431, which needs re-writing for clarity.

Reference to microglia infected by SARS-CoV needs a citation (lines 523-525).

Experimental design

The article content is within the Aims and Scope of the journal, using a tried and tested systematic review approach to evaluate the current state of the art in the literature. The Survey Methodology is consistent with a comprehensive, unbiased coverage of the subject matter. Sources are adequately cited, with some queries raised above. The review is organized logically.

Validity of the findings

No data are presented with which to comment on, as this is a review.

Additional comments

Nothing to add.

Reviewer 2 ·

Basic reporting

The current systematic review and meta-analysis of works evaluating possible potential link of neuroinflammation alterations during PACS to adult neurogenesis.

The manuscript is comprehensive, well written, and follows appropriate PRISMA guidelines for a systematic review.

Minor comments:

- Abbreviations should be defined the first time they are used in the text.

Please revise there are several abbreviations not defined at first instance where they are used.

Eg: spell out ‘RAS’ in sentence 290 the first time it is used, instead in sentence 408.

- Please revise ‘sentence 488’ there is some ambiguity


Please see detailed comments in the annotated manuscript

Experimental design

- The literature screening was conducted according to the standard methods and authors clearly stated how they followed the PRISMA guidelines. The authors searched Pubmed, Cochrane, Scopus, Web of Science, Google Scholar and ResearchGate.

- It provides a useful quantitative analysis of the current research data and also evaluates potential theories on underlying mechanisms.

Validity of the findings

- The review provides evidence showing effects of adult neurogenesis as potential cause to neuropsychiatric symptoms/disorders in PACS.

- The discussion is well written with different theoretical models and discusses the limitations of this systematic review.

Annotated reviews are not available for download in order to protect the identity of reviewers who chose to remain anonymous.

---

## Round 0.2 · accepted · Accept

The rebuttal and track changes have been studied, cross checked and revisions noted. It is currently in an acceptable format.